# Defect Detection of Pipeline Inner Surface Based on Coaxial Digital Image Correlation with Hypercentric Lens

**DOI:** 10.3390/ma15217543

**Published:** 2022-10-27

**Authors:** Jiankang Qi, Mengqiao Xu, Weiling Zhang, Yubo Liu, Xiangjun Dai

**Affiliations:** School of Transportation and Vehicle Engineering, Shandong University of Technology, Zibo 255049, China

**Keywords:** defect detection, pipeline, hypercentric lens, digital image correlation

## Abstract

A coaxial dual-camera digital image correlation system using a hypercentric lens was proposed to determine the defect position in the inner wall of a pipeline under loads. Compared with the traditional dual-camera system, this system ensures that both cameras can capture a 360-degree panoramic image in the same position. Herein, the imaging principle of the system was introduced in detail. In addition, the effectiveness and accuracy of the proposed method were verified through verification and application experiments.

## 1. Introduction

Currently, pipeline structures are widely used for transporting various corrosive, toxic, flammable, and explosive liquids and gases. For such structures, small defects inevitably appear during the manufacturing process. Moreover, during the service period, when the pipeline is under a load, the defect can cause damage or fracture and spread from the inside to the outside, resulting in pipeline leakage. Consequently, this could lead to explosion accidents and huge losses. Therefore, the detection of defects in pipelines is particularly important.

In recent years, in the context of large-scale industries, non-destructive testing has been crucial in detecting defects and physical parameters, which to a certain extent, reflect the level of industrial development in a country. Currently, the non-destructive testing methods for evaluating the inner wall of a pipeline include magnetic flux leakage testing, eddy current, ultrasonic, and radiographic testing techniques [1]. In magnetic flux leakage detection, a strong magnet on the detector generates an axial magnetic field on the inner wall of the pipe. Therefore, if the pipe wall is damaged, the magnetic field changes [2,3,4]. Eddy’s current testing is based on the principle of electromagnetic induction. Notably, pipeline defects cause changes in eddy current impedance, and therefore, the defects can be detected using eddy current testing instruments [5,6]. In ultrasonic technology, an ultrasonic wave propagates in a medium and is received by a probe after reflection. The shapes and sizes of the defects and other parameters are calculated by reflection from different angles [7,8,9,10]. Radiographic detection uses the attenuation law of radiation in the process of penetrating objects to detect defects in the inner walls of pipes [11,12]. However, the limitations of the aforementioned methods restrict their applications in engineering fields. For example, a magnetic flux–leakage–detection probe is easily affected by a pipe wall, resulting in insufficient data. Moreover, it is more applicable for ferromagnetic materials with a simple shape. Ultrasonic detection designs are complicated and expensive, and the strong signals in the eddy current technology interact with each other. Among these limitations, interference, fast energy consumption, complex radiation-technology equipment, high cost, and radiation hazards are the most critical. Currently, in the field of experiments, techniques such as endoscopy [13] and electronic speckle interferometry [14] are used in non-destructive testing of the inner walls of pipes.

Among the various non-destructive testing techniques, digital image correlation (DIC) is a visualization technique for shape and deformation measurement with unique advantages [15,16,17,18]. Compared with other measurement methods, this method has the advantages of low environmental vulnerability and simple sample preparation. To obtain a full-field displacement distribution and eliminate the interaction between in-plane and out-of-plane displacements, a 3D DIC method was developed based on the principle of binocular stereo vision [19,20]. Traditional 3D DIC systems include two cameras to capture the same scene from different directions to extract displacement information and reconstruct 3D shapes. However, the traditional 3D DIC technology is extremely limited for pipeline measurements. First, ordinary cameras cannot observe 360-degree images of the inner wall of the pipeline. Secondly, two cameras cannot capture the exact same area. To address these issues, a single-camera 3D DIC system has been proposed [21,22,23]. The single-camera field of view is divided into two by a prism, thereby addressing the aforementioned limitation. However, as the field of view decreases, a reduction in resolution increases the error. Additionally, specular reflection combined with 3D DIC has been used to measure the inner wall of a pipe by reflecting the pipe image [24,25]. In our previous work, a dual biprism stereo camera 3D DIC system combined with a strain-based method was applied to detect cracks on the internal surface [26]. However, the disadvantage of this kind of method is that the entire circumference of the pipe cannot be measured simultaneously.

To address the aforementioned challenges, this study proposes a coaxial dual-camera system using a hypercentric lens to determine the defect position of a pipeline under loads. Compared with the traditional dual-camera DIC system, this system ensures that both cameras can capture a 360-degree panoramic image at the same position. In addition, the system is suitable for testing narrow pipelines. Herein, the imaging principle of the system is introduced in detail, and verification and compression experiments are described to verify the effectiveness, practicability, and accuracy of the proposed system.

## 2. Methods

### 2.1. Coaxial Dual-Camera System with Hypercentric Lens

Figure 1a,b shows the schematic of the imaging optical paths of the traditional dual-camera measurement system and the coaxial dual-camera measurement system, respectively. As seen in Figure 1, although the traditional dual-camera measurement system can capture the image of the specimen, when measuring objects as narrow as a pipe or with a large curvature, determining whether the axis of the lens is parallel or at an angle to the same area using images is challenging. In Figure 1a, the observation areas of the left and right cameras are noticeably different, and the images collected on the computer are the morphologies of both sides of the inner wall of the pipeline, which cannot be matched.

The coaxial dual-camera system is shown in Figure 1b, where camera A is in front of the shooting area, and a semi-reflective and semi-transparent beam-splitting prism is placed between the object and the camera to be measured. The positions are at an angle of 90°. To ensure consistent image formation, the distances from the two cameras to the prism are the same, as are the focal length and aperture of the lenses. When the light path at the bottom of the pipe passes through the beam-splitting prism, half of the light intensity enters the two cameras. At this time, the images captured by cameras B and A are in a mirror–image relationship. Then, the image of camera B is mirrored horizontally to obtain the same image as camera A. If the coaxial system in Figure 1b is directly used for observation, defect detection can be performed for the part of the inner wall pipeline. However, the information of a complete 360-degree inner cylindrical surface cannot be achieved, resulting in inaccurate final results. Therefore, this study uses a single hypercentric lens that can focus on the inner wall and bottom surface of the object simultaneously, as shown in Figure 2.

Figure 3 shows the model of a hypercentric lens. The aperture stop is placed behind the image-side focal point so that the convergence point is in object space [27]. If an object (such as Object 1 in Figure 2) is placed between the convergence point and the lens, the object appears smaller in the image when it is closer to the lens, while the object appears larger if it (such as Object 2 in Figure 2) is placed before the entrance pupil appears larger. These are the two observation modes of the hypercentric lens: exterior and inner wall measurements. When the measured object is located in front of the convergence point, the outer wall of the object can be observed. However, when it is located behind, the inner wall and bottom surface can be observed. The latter one was applied in this study.

Before the experiment, a single hypercentric lens, which can capture real-time images of the pipeline, was placed in front of the beam-splitting prism, as shown in Figure 4. Cameras A and B are German IDS UI-1540LE-M-GL industrial cameras with an EDMUND model 50 mm focal length lens. Both cameras were moved to obtain perfectly horizontal mirror images, ensuring that they could see the area in the lens. A blue ring light source was used to provide uniform light throughout the pipe. Figure 5a,b compares the field of view of the common coaxial system and the system using the hypercentric lens, respectively.

### 2.2. Three-Dimensional Digital Image Correlation

The binocular stereo-vision effect of this system is shown in Figure 6. Camera A, camera B, and the measured object are in different coordinate systems, and the images are in a mirror–image relationship. *O*_W_ is the origin of the world coordinate system, and *O*_A_ and *O*_B_ are the camera optical centers of two cameras, respectively. In contrast to the traditional 3D DIC system, this system first needs to mirror the image from camera B, and the direction of the AB image is the same at this time. Before the experiment started, both cameras were calibrated to determine their internal and external parameters and eliminate errors. Table 1 and Table 2 show the calibration data for the internal and external parameters.

Calibration parameters can be classified into two groups: internal and external parameters. Internal parameters include the camera’s main point, focal length, and distortion. External parameters include rotational and translational vectors between the cameras. In actual measurements, the internal parameters are generally the inherent properties of the camera and do not change during the experiment. The external parameters vary with the position of the camera. Therefore, if the camera position changes, it should be recalibrated.

After the calibration, two images were captured at different times, and the same physical points were determined using a correlation matching algorithm. Figure 7(a1,b1) are the images before deformation captured by the left and right cameras, respectively, and Figure 7(a2,b2) are the images after deformation. The displacement of the region of interest (ROI) is calculated using a correlation-matching criterion for the image before and after deformation. The zero-mean normalized cross-correlation criterion (ZNCC) is one of the most widely used criteria. The mathematical function of ZNCC can be defined as follows:(1)$$C=\frac{\sum_{x=-N}^{N}\sum_{y=-N}^{N}\left[f\left(x,y\right)-\bar{f}\left(x,y\right)\right]\left[g\left(x^{\prime },y^{\prime }\right)-\bar{g}\left(x^{\prime },y^{\prime }\right)\right]}{\sqrt{\sum_{x=-N}^{N}\sum_{y=-N}^{N}{\left[f\left(x,y\right)-\bar{f}\left(x,y\right)\right]}^{2}}\sqrt{\sum_{x=-N}^{N}\sum_{y=-N}^{N}{\left[g\left(x^{\prime },y^{\prime }\right)-\bar{g}\left(x^{\prime },y^{\prime }\right)\right]}^{2}}}\;$$
(2)$$\;\;\;\;\bar{f}\left(x,y\right)=\frac{1}{2N+1^{2}}\sum_{x=-N}^{N}\sum_{y=-N}^{N}f\left(x,y\right),\;$$
(3)$$\bar{g}\left(x^{\prime },y^{\prime }\right)=\frac{1}{2N+1^{2}}\sum_{x=-N}^{N}\sum_{y=-N}^{N}g\left(x^{\prime },y^{\prime }\right),\;$$
where $f\left(x,y\right)$ is the gray value at the reference image point $\left(x,y\right)$, and $g\left(x^{\prime },y^{\prime }\right)$ is that at the target image point $\left(x^{\prime },y^{\prime }\right)$. Additionally, $\bar{f}\left(x,y\right)$ is the average intensity value of the reference subset, and $\bar{g}\left(x^{\prime },y^{\prime }\right)$ is that of the target subset. N is the subset size used for computation. Then, the sub-pixel displacement is solved using the forward Gauss–Newton method (FA-GN) and the inverse Gauss–Newton method (IC-GN), and the specified calculation points can be stably matched to the target point. Finally, to determine the correspondence of the calculated points in all the images, triangulation is used to reconstruct the 3D coordinates. By comparing the 3D coordinates of all the points before and after deformation, the full-field displacement and strain distribution after loading are obtained.

## 3. Experimental Results

### 3.1. Shape Reconstruction of the Inner Surface

A PVC pipeline has good corrosion resistance and water tightness; usually, a pipeline through which liquid can pass with small fluid resistance is used in the discharge of industrial sewage and building construction. It is made of PVC, adhesive, and stabilizer with extrusion molding. In this experiment, a round PVC pipe with an outer diameter of 40 mm, a wall thickness of 3 mm, and a length of 90 mm was taken as the research object. Before the experiment, black and white speckles were sprayed on the inner surface of the pipe as carriers of deformation information.

In this study, the feasibility and accuracy of the experiment were determined using topography recovery and pipe inner diameter measurements at different depths. Figure 8a shows the calculation area selected for the experiment, and the 3D contour map of the inner surface (Figure 8b) is obtained through subsequent DIC-related calculations. The depth of the measured area in the z direction is approximately 38 mm.

According to the results in Figure 8, the inner diameter of the pipeline at different depths is compared with the actual inner diameter of the pipeline, and the results are as follows. Figure 9a,b represent the four line segments selected in the *x*- and *y*-directions when the depth of the calculation area is 24 mm and 30 mm, respectively. The comparison is presented in Table 3. The results indicate that the relative error between the measured and real values is less than 1%. Thus, the system can be applied to the defect detection of the inner wall of the pipeline.

### 3.2. Defect Detection

In this study, pipelines with penetrating gaps were tested for the presence of defects. The experimental setup is shown in Figure 10. Two cameras were positioned at an angle of 90° near to the beam splitter. A hypercentric lens was placed between the beam splitter and the pipeline. The optical axis of the hypercentric lens was set to coincide with the optical axis of camera B. A blue ring light source was used to illuminate the pipeline’s inner surface. Before the experiment, the distance between the hypercentric lens and the pipe was changed so that the camera presented a clear and complete image. In the experiment, the pipeline was compressed, the camera captured images of the inner wall of the pipeline before and after deformation in real-time, and the position of the defect was determined using the strain distribution of the deformed image.

The pipe size and material used in the compression process were the same as those in the previous section. However, in this experiment, a transverse notch on the side wall was prefabricated for the pipe, and the notch size was 2 cm × 0.5 cm. During the experiment, the two industrial cameras were used to capture images inside the pipeline. After calibrating the camera using an 11 × 8 checkerboard calibration board, the strain was calculated using the DIC method.

Figure 11a,b shows the internal images of the pipeline captured by cameras A and B of the co-axial 3D system. The yellow area in Figure 11a represents the inner wall of the pipeline. This wall is also the calculation area for this experiment. Generally, the quality of sprayed speckle particles should be evaluated before the experiment. A main quality assessment parameter of speckle patterns is the speckle size. One approach for characterizing the feature size is to employ a form of full width at half maxima of the auto-correlation function [18]. In addition, the correlation coefficient is defined as:(4)$$A=\frac{\sum_{i=1}^{M}\sum_{j=1}^{N}I\left(x_{i},y_{j}\right)I\left(x_{i}-u,y_{j}-v\right)}{\sum_{i=1}^{M}\sum_{j=1}^{N}{\left[I\left(x_{i},y_{j}\right)\right]}^{2}}$$
where $I\left(x_{i},y_{j}\right)$ denotes the image intensity at position $\left(x_{i},y_{j}\right)$ and *u* and *v* are translations along the *x* and *y* axes, respectively.

Figure 11c–e show the texture analysis results of the randomly selected region in the image subregion. It can be seen that the selected area has a particularly noticeable peak value. The full-width value of the vertical half-peak of the peak was 6.5 pixels, and the full-width value of the horizontal half-peak was 5 pixels. This indicates that the speckle particles are of good quality and can be used for experimental measurements.

Figure 12 shows the displacement and strain distribution of the inner wall at 10 kN, and region A is the defective part. When the pipeline is compressed, a discontinuity occurs in the displacement distribution of the defect in the *x*-direction, and the position of the notch can be determined approximately. In the calculation results of principal strain in the *x*-direction, a strain concentration area is generated in the defect region, wherein the absolute value of strain is significantly higher than that in other parts of the region of interest. Therefore, the specific position of the notch can be determined according to the displacement and strain information.

Figure 13 shows the strain distribution of pipe defects under compression at different positions, where Figure 13a–c represent the positions of defects on the top, right, and bottom, respectively. Notably, the strain at other locations is uniform, whereas a noticeable strain concentration is observed at the defect location. Therefore, the co-axial dual-camera 3D digital image correlation system can determine the specific locations of defects in the inner wall of the pipeline.

## 4. Conclusions

In this study, a dual-camera co-axial 3D measurement system was developed. The system consists of two industrial CCD cameras, a semi-reflective and semi-transparent beam-splitting prism, and a hypercentric lens. The beam-splitting prism allowed two cameras to observe the same field of view, and a hypercentric lens was placed in front of the beam-splitting prism. As such, one lens acts on two cameras, significantly reducing the cost.

The system solves the space limitation challenge of the traditional dual-camera system and the low image resolution of the single-camera 3D system. The effectiveness and accuracy of the 3D-DIC method were verified through verification and application experiments. The experimental results showed that, compared with the traditional 3D-DIC, the system could observe the strain concentration of the defects at different positions of the pipeline. Therefore, the location of the defects could accurately be determined. In addition, a panoramic internal image of the inner wall of the pipeline can be obtained simultaneously. The system could also measure the strain of the defection-free pipeline under different loads. Interestingly, the system structure was simple and easy to debug. However, because of the limited depth of field range of industrial cameras, the acquired image depth cannot achieve the desired effect, and the side of the pipe wall away from the camera will appear jagged, resulting in an incomplete calculation area. In addition, if the specimen diameter is small, the camera will not be able to capture a clear image. Although the annular light source can uniformly illuminate most areas, those far away or near the light source will still be extremely dark or exposed.

## Figures and Tables

**Figure 1 materials-15-07543-f001:**
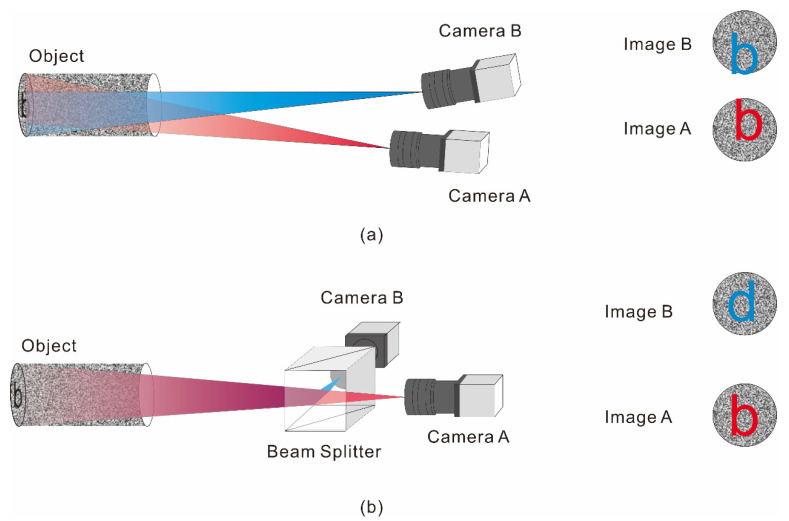
Optical path comparison. (**a**) Optical path of traditional dual-camera 3D measurement system; (**b**) Optical path of the coaxial dual-camera 3D measurement system.

**Figure 2 materials-15-07543-f002:**
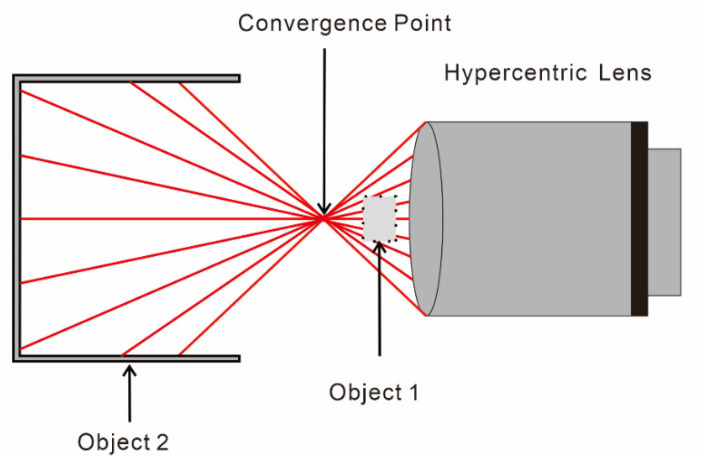
Principle of the hypercentric lens.

**Figure 3 materials-15-07543-f003:**
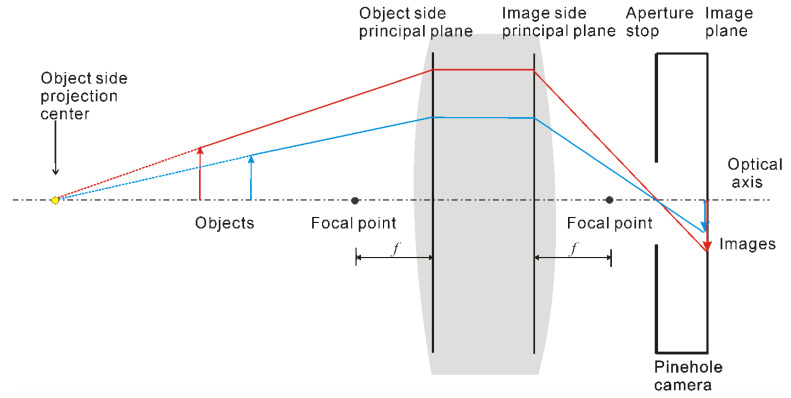
Model of a hypercentric lens.

**Figure 4 materials-15-07543-f004:**
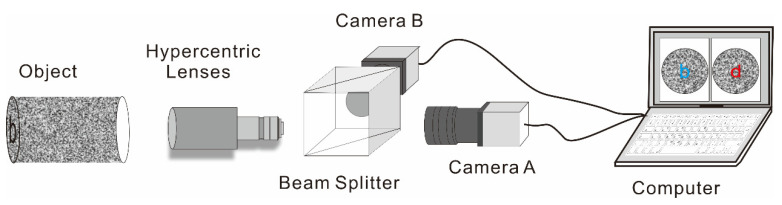
Schematic diagram of the coaxial dual camera with hypercentric lens.

**Figure 5 materials-15-07543-f005:**
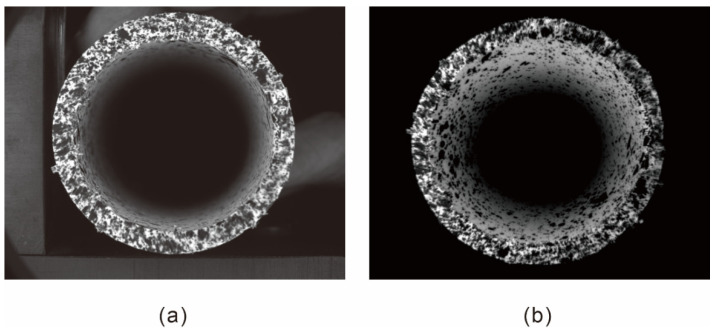
Comparison of field of view (**a**) without hypercentric lens and (**b**) with hypercentric lens.

**Figure 6 materials-15-07543-f006:**
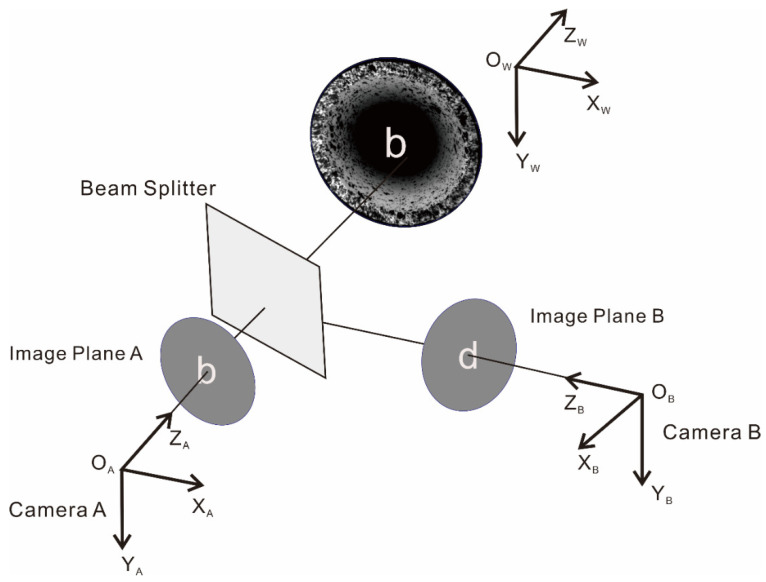
Schematic of binocular stereo visual effect of coaxial dual-camera system.

**Figure 7 materials-15-07543-f007:**
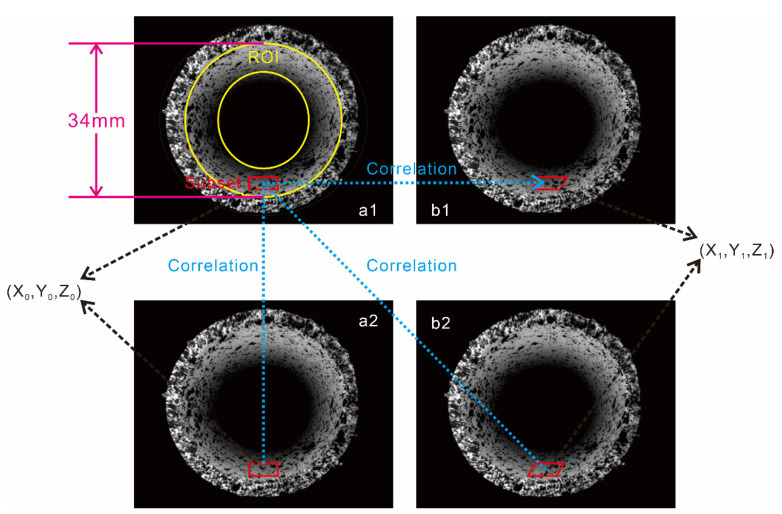
Principle of stereo matching of 3D-DIC images. (**a1**,**a2**) left image before and after deformation; (**b1**,**b2**) right image before and after deformation.

**Figure 8 materials-15-07543-f008:**
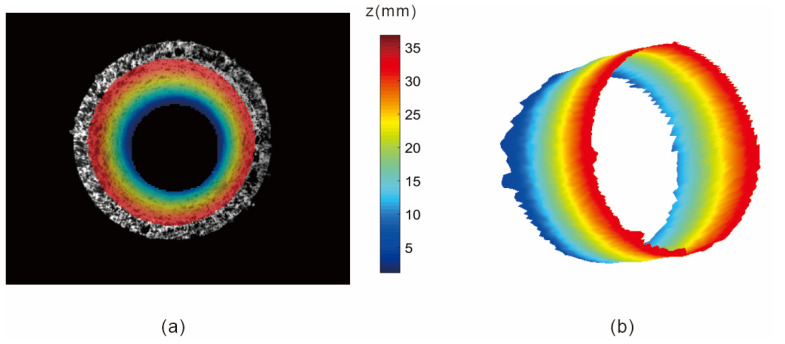
(**a**) Real calculation area; (**b**) Three-dimensional outline of the inner surface.

**Figure 9 materials-15-07543-f009:**
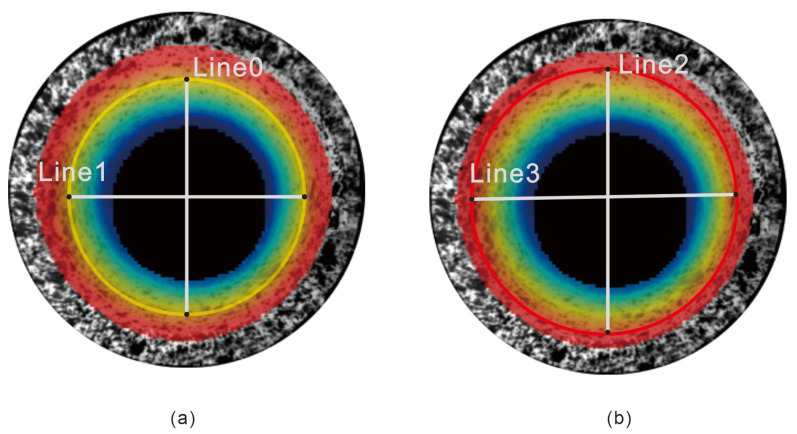
Random diameters at different depths: (**a**) 24 mm; (**b**) 30 mm.

**Figure 10 materials-15-07543-f010:**
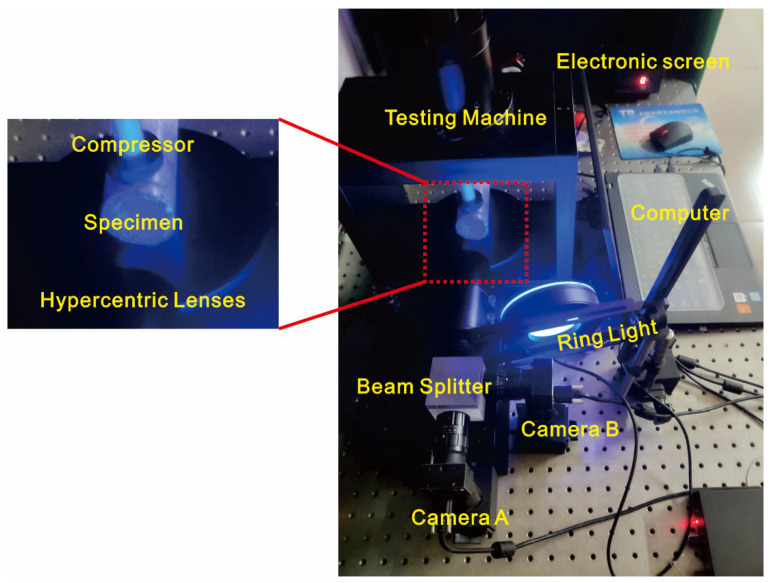
Pipeline compression test.

**Figure 11 materials-15-07543-f011:**
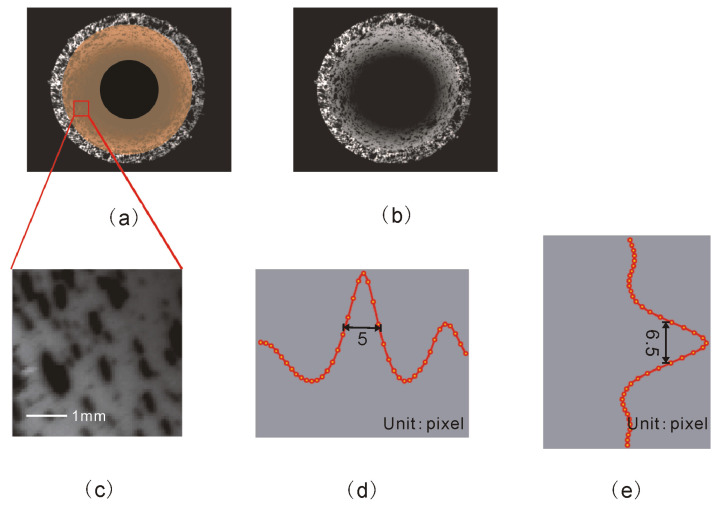
Collected images and speckle texture analysis: (**a**) image of camera A, (**b**) image of camera B, (**c**) typical areas of the subregion, (**d**) full width of horizontal half-peak, and (**e**) full width of vertical half-peak.

**Figure 12 materials-15-07543-f012:**

Pipe wall displacement and strain distribution under 10 kN load: (**a**) displacement of *u*-field; (**b**) displacement of *v*-field; (**c**) $\varepsilon_{xx}$ strain.

**Figure 13 materials-15-07543-f013:**
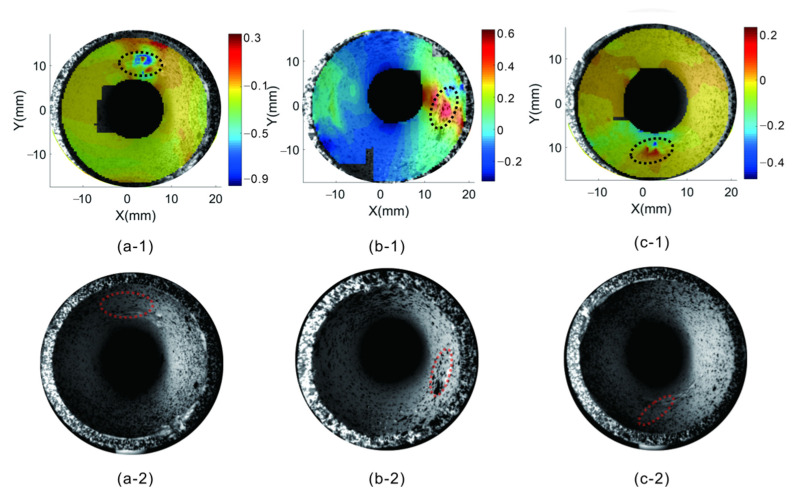
Strain distribution of defects at different positions: (**a**) upper part; (**b**) on the right side; (**c**) bottom.

**Table 1 materials-15-07543-t001:** Internal parameter calibration data.

Internal Parameters	Camera A	Camera B
Center x (pixels)	1013.62	1046.31
Center y (pixels)	535.97	505.16
Focal length x (pixels)	5054.18	4967.7
Focal length y (pixels)	5037.47	4958.05
Distortion	−0.0978	−0.19

**Table 2 materials-15-07543-t002:** External parameter calibration data.

External Parameter	*x*	*y*	*z*
Rotating vector (degrees)	−1.84563	−6.24523	0.677195
Translation vector (mm)	19.526	−1.04709	−3.78366

**Table 3 materials-15-07543-t003:** Error between the measured and actual data of the inner diameter of the pipe.

	Measured Data	Real Data	Absolute Error (mm)	Relative Error (%)
Line 0	33.67	34.00	0.33	0.97
Line 1	33.80	34.00	0.20	0.59
Line 2	33.95	34.00	0.05	0.15
Line 3	34.28	34.00	0.28	0.82

## Data Availability

Not applicable.

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
