# Peer review of "Defect Detection of Pipeline Inner Surface Based on Coaxial Digital Image Correlation with Hypercentric Lens"

_materials, 2022, doi:10.3390/ma15217543_

Round 1
Reviewer 1 Report
The manuscript presents an application of the DIC method for quality control of PVC pipelines. A coaxial dual-camera system with a hypercentric lens was proposed to detect positions of defects in the inner wall of a pipeline under load. The system is capable to capture a 360-degree panoramic image of the pipeline wall. The effectiveness and accuracy of the proposed method were experimentally verified.
In the attached file, comments and suggestions for the Authors can be found.

Reviewer 2 Report
In this paper, the authors proposed a coaxial digital image correlation method with hypercentric lens for detecting the defect of pipeline's inner surface. Although the results may be of interest to some readers, there are some concerns that must be clarified:
1. The author mentioned that the images of the two cameras obtained by the beam splitter are mirror images, which is equivalent to the images captured by the two cameras at an angle of 180°. However, according to the measurement principle of the binocular camera, if the angle between the two cameras is 180°, the measurement of the depth direction cannot be completed. Please explain in detail how the setup completes the stereo DIC measurement.
2. The author used a hypercentric lens to capture the image of the inner wall of the pipeline. Does this lens have image distortion, and if so, how to correct it?
3. Since the region of interest is the inner wall of the pipeline, its 360° range will inevitably bring inconvenience to the calibration process. Please give details on how to solve the difficulties in calibration.
4. The author mentioned that an autocorrelation half-width method was used to evaluate the quality of sprayed speckle particles. What is the autocorrelation half-width method, and what do the peaks in Figure 10(d), (e) represent?
Reviewer 3 Report
Dear authors,
it is a exciting topic. I wrote a few comments regarding your publication. See the attached file.

Round 2
Reviewer 2 Report
The authors have adequately taken into account the points made in the previous report.